# In Search of the Holy Grail: Toward a Unified Hypothesis on Mitochondrial Dysfunction in Age-Related Diseases

**DOI:** 10.3390/cells11121906

**Published:** 2022-06-12

**Authors:** Jun Zhang, Yuguang Shi

**Affiliations:** Department of Pharmacology, Sam and Ann Barshop Institute for Longevity and Aging Studies, University of Texas Health Science Center at San Antonio, San Antonio, TX 78229, USA; zhangj7@uthscsa.edu

**Keywords:** ALCAT1, cardiolipin, mitochondrial dysfunction, aging, age-related diseases

## Abstract

Cardiolipin (CL) is a mitochondrial signature phospholipid that plays a pivotal role in mitochondrial dynamics, membrane structure, oxidative phosphorylation, mtDNA bioenergetics, and mitophagy. The depletion or abnormal acyl composition of CL causes mitochondrial dysfunction, which is implicated in the pathogenesis of aging and age-related disorders. However, the molecular mechanisms by which mitochondrial dysfunction causes age-related diseases remain poorly understood. Recent development in the field has identified acyl-CoA:lysocardiolipin acyltransferase 1 (ALCAT1), an acyltransferase upregulated by oxidative stress, as a key enzyme that promotes mitochondrial dysfunction in age-related diseases. ALCAT1 catalyzes CL remodeling with very-long-chain polyunsaturated fatty acids, such as docosahexaenoic acid (DHA). Enrichment of DHA renders CL highly sensitive to oxidative damage by reactive oxygen species (ROS). Oxidized CL becomes a new source of ROS in the form of lipid peroxides, leading to a vicious cycle of oxidative stress, CL depletion, and mitochondrial dysfunction. Consequently, ablation or the pharmacological inhibition of ALCAT1 have been shown to mitigate obesity, type 2 diabetes, heart failure, cardiomyopathy, fatty liver diseases, neurodegenerative diseases, and cancer. The findings suggest that age-related disorders are one disease (aging) manifested by different mitochondrion-sensitive tissues, and therefore should be treated as one disease. This review will discuss a unified hypothesis on CL remodeling by ALCAT1 as the common denominator of mitochondrial dysfunction, linking mitochondrial dysfunction to the development of age-related diseases.

## 1. Introduction

Despite the rapid progress in biomedical research in recent years, incidences of age-related chronic diseases, such as obesity, cardiovascular diseases, type 2 diabetes (T2DM), Alzheimer’s disease, and cancer are on the rise. However, the underlying causes of these disorders remain poorly understood, which prevents the development of effective treatments for these conditions. Consequently, most of the current treatments for age-related diseases are limited to relieving the symptoms. Moreover, there is no effective treatment for some of the most devastating diseases, such as heart failure, diabetic complications, strokes, and Alzheimer’s disease. Although it is commonly believed that genetic predisposition plays a major role in the development of these chronic disorders, no single gene has been identified as the underlying cause for these chronic conditions to date. On the other hand, there is a close association of obesity with the development of all the metabolic diseases [1,2]. Since our genomic polymorphism and genetic selection could not have evolved in such short period of time, there is a major gap in our understanding of how environmental factors such as over-nutrition contribute to the development of age-related metabolic diseases.

Mitochondria are ancient bacterial symbionts that are tamed by the eukaryotic host cells. However, they retain their own DNAs (mtDNA) which do not follow the same rule of replication as genomic DNA. As matter of fact, many of the mitochondrial behaviors, such as fusion and fission, are totally new inventions by the eukaryotic cells. Additionally, mitochondria lack the SNARE complex, therefore rely upon membrane contacts for the transport of mitochondrial proteins and lipids [3,4,5,6]. Furthermore, the number of mitochondria varies per cell, ranging from a few hundred to a few thousand, which renders them different from Mendelian inheritance. As they provide energy by burning the calories in our diet, mitochondrial respiration also generates most of the endogenous ROS, which causes damage to the mitochondrial membrane proteins, mtDNA, and phospholipids, such as CL. This cumulative damage functions as an aging clock, as proposed by Dr. Harman more than 65 years ago [7,8], triggering the onset of age-related diseases when the cumulative damages reach the threshold. Hence, the mitochondrial dysfunction associated with aging embodies all the features necessary to explain the observed characteristics of the common age-related diseases [9] and has been proposed as the missing link between aging and the development of age-related diseases [2].

CL is a polyglycerophospholipid exclusively localized in the mitochondria, where it regulates membrane structure, oxidative phosphorylation, mtDNA biogenesis, fusion, fission, and mitophagy in eukaryotic cells [10,11,12,13,14]. This role is mediated in part by the acyl composition of its four side chains, which are dominated by linoleic acid in insulin-sensitive tissues [15,16,17,18]. This unique acyl composition is not derived from the de novo synthesis of CL but rather from a remodeling process that involves phospholipases and acyltransferases/transacylases [19,20,21]. This remodeling process is also believed to be responsible for generating CL species that are highly sensitive to oxidative damage by ROS, further exacerbating CL peroxidation and oxidative stress. In this review, we make the case that the pathological remodeling of CL by ALCAT1 in response to oxidative stress associated with aging and obesity is the missing link between mitochondrial dysfunction and the development of various age-related diseases. We will also review evidence that age-related disorders are one disease (aging) manifested by different mitochondrion-sensitive tissues, and therefore can potentially be treated as one disease in the future.

## 2. CL Remodeling and Acyl Composition

CL is the only phospholipid with four acyl chains, which, in theory, can generate millions of CL species (1 × 4^20–40^) (Figure 1A). However, the four fatty acyl chains of CL are restricted to C18 chains dominated by the linoleoyl group (C18:2) in the heart, skeletal muscle, and liver [16,17,18,22]. This unique fatty acyl composition, also known as tetralinoleoyl CL (TLCL, Figure 1B), plays an important role in the maximum efficiency of respiration. The hydrophobic double-unsaturated linoleic diacylglycerol species is believed to be required for the high-affinity binding of CL to proteins, whereas TLCL is required for mitochondrial membrane structure in the heart [23]. Thus, an alteration in the molecular species composition of CL affects the activities of cytochrome *c* oxidase and other electron transport chain enzymes [24,25]. However, the formation of this unique fatty acyl composition of CL does not occur during de novo biosynthesis because the enzymes of the CL biosynthetic pathway lack appropriate substrate selectivity [26,27,28]. This is further confirmed by the studies on human CL synthase [29,30,31]. Thus, newly synthesized CL is believed to undergo a remodeling process to achieve its appropriate acyl content.

Two distinct mechanisms have been posited to carry out the CL remodeling process in mammals (Figure 2) [15,32,33,34]. The first mechanism involves transacylation of acyl groups from phosphatidylcholine (PC) or phosphatidylethanolamine (PE) to CL, which is partly catalyzed by tafazzin (TAZ), a transacylase enzyme that when mutated causes defective CL remodeling and Barth syndrome (BTHS) [19]. The alternative pathway involves deacylation by phospholipase A2 to lysocardiolipin, followed by reacylation to CL by two acyl-CoA-dependent lysocardiolipin acyltransferases (Figure 2). The first acyltransferase is the acyl-CoA:lysocardiolipin acyltransferase (ALCAT1) which catalyzes the remodeling of CL with both monolysocardiolipin (MLCL) and dilysocardiolipin as substrates at the mitochondrial associated membrane [18,20]. The second acyltransferase is an MLCL acyltransferase (MLCLAT), which catalyzes the synthesis of CL with MLCL as the substrate [35]. Intriguingly, MLCLAT shares the same sequence homology with the mitochondrial trifunctional protein, which is localized at the inner mitochondrial membrane [21,36]. In comparison to the MLCLAT, which catalyzes exclusively the synthesis of TLCL by using linoleoyl-CoA and MLCL as substrates, ALCAT1 lacks a preference for linoleic acid as a substrate [20,21]. Consequently, CL remodeling by ALCAT1 with very-long-chain fatty acids, such as docosahexaenoic acid (DHA) and arachidonic acid, leads to TLCL deficiency, oxidative stress, and mitochondrial dysfunction, which is implicated in the pathogenesis of various age-related diseases [17,18,37,38,39].

## 3. CL Acyl Composition, Oxidative Stress, Mitochondrial Fragmentation, and Aging

Mitochondria are the only organelles in eukaryotic cells that possess double membranes. The inner mitochondrial membrane is with high degree of curvature and enriched with CL which functions as the “glue” for mitochondrial respiration complexes. The inner membrane is the most protein-condensed membrane in the cell, which is believed to be essential for the proper function of the electron transport chain [40]. This unique membrane structure is made possible by the presence of CL. CL is a cone-shaped non-bilayer phospholipid predominantly localized in the curved membrane sites of mitochondrial cristae, where it is believed to stabilize respiratory chain complexes (Figure 3) [41,42,43]. Consequently, CL depletion in age-related diseases leads to mitochondrial fragmentation, oxidative stress, defective oxidative respiration, and mtDNA release to the cytosol (Figure 3) [17,18,37,38,44,45,46,47]. CL is particularly sensitive to the oxidation of its double bonds by ROS due to its exclusive location near the site of ROS production in the inner mitochondrial membrane [48]. The mitochondrial electron transport chain is considered a major intracellular source of ROS production both during physiologic respiration and during disrupted electron transport [25,48]. CL oxidation produces lipid peroxides which are trapped in the mitochondria, and therefore are far more toxic than other forms of ROS. In addition, CL peroxides further damage neighboring acyl chains, triggering a vicious cycle known as CL peroxidation, oxidative stress, and mitochondrial dysfunction. Moreover, CL peroxidation by ROS disrupts its binding with cytochrome *c* and affects the activity of complexes I, III, and IV of the mitochondrial respiratory chain [49]. A burst of ROS damages mitochondria by causing profound loss of CL [50]. Furthermore, CL deficiency in ischemia and reperfusion results in mitochondrial dysfunction, leading to a decreased oxidative capacity, the loss of cytochrome *c*, and the generation of ROS. CL, but not its peroxidized form, is able to almost completely restore the ROS-induced loss of cytochrome *c* oxidase activity [51]. In support of the key role of CL peroxidation in mitochondrial dysfunction, CL is the only phospholipid that undergoes oxidation during the onset of apoptosis [52].

The oxidative injury of the mitochondria impacts critical aspects of the aging process and contributes to impaired physiological function, which has been proposed as the primary cause of aging. In support of the causative role of oxidative stress in aging, ROS level and phospholipid peroxidation index are inversely correlated with lifespan in mice and humans [53,54,55]. Oxidative stress is also believed to contribute to an age-associated decline in CL. Consistent with an increased level of ROS, aging is associated with CL deficiency and the profound remodeling of CL, similar to that observed in metabolic diseases. Aging and physical exercise reciprocally affect mitochondrial and cardiac function by regulating CL levels in the heart. Exercise increases insulin sensitivity and the level of TLCL, whereas aging decreases TLCL level with a concurrent increase in the long-chain polyunsaturated fatty acids (PUFA) content in CL [16,56,57]. Aging also decreases CL content in the heart, liver, and kidney, leading to the decreased activity of the mitochondrial phosphate transporter, pyruvate carrier, adenine nucleotide transporter, and cytochrome oxidase, all of which require CL for optimum activity [58,59,60,61,62]. These defects can be restored by the supplementation of acyl-carnitine, which is believed to restore CL levels [12].

## 4. ALCAT1 Controls Mitochondrial Etiology of Age-Related Diseases

ALCAT1 is the first acyl-CoA dependent lysocardiolipin acyltransferase previously identified by us [20]. Our recent studies show that ALCAT1 catalyzes the pathological remodeling of CL with aberrant acyl compositions commonly found in age-related diseases, including enrichment of DHA in CL, leading to TLCL depletion and mitochondrial dysfunction [18]. Moreover, ALCAT1 expression is upregulated by oxidative stress associated with age-related diseases, which is implicated in the mitochondrial etiology of various age-related diseases in mice. Accordingly, the overexpression of ALCAT1 causes severe oxidative stress, mitochondrial swelling, and defective oxidative phosphorylation, both in metabolic cell lines and in multiple mouse models of age-related metabolic diseases [17,18,47]. Consequently, the targeted deletion of the *ALCAT1* gene in mice not only prevented mitochondrial dysfunction, but also the development of obesity, T2DM, fatty liver disease, Parkinson’s disease, and cardiomyopathy [18,38,47,63]. Moreover, ALCAT1 is primarily expressed in vascular endothelial cells, where it functions as key mediator of hypoxic response, and therefore, upregulated ALCAT1 expression is also implicated in the pathogenesis of heart failure and cancer [17,64,65].

### 4.1. Diabetes and Obesity

Obesity and T2DM are characterized by systemic oxidative stress, which is believed to be a principal causative factor of insulin resistance and other obesity-related metabolic complications [66,67,68,69,70]. Diabetes and obesity also cause a CL deficiency and the profound remodeling of CL’s acyl composition [15,18,22]. In support of the causative role of ALCAT1 in the pathogenesis of T2DM and obesity, *ALCAT1* mRNA levels were upregulated by oxidative stress and diet-induced obesity (DIO). The stable overexpression of ALCAT1 in C2C12 skeletal myocytes led to CL depletion, the enrichment of DHA content in CL, severe oxidative stress, mitochondrial swelling, and a loss of cisterna structure (Figure 4) [18]. Consequently, the targeted deletion of the *ALCAT1* gene not only prevented the onset of DIO but also insulin resistance in metabolic tissues. ALCAT1 deficiency also significantly increased TLCL content in the heart, leading to significant improvement in mitochondrial complex I activity [18]. Additionally, ablation of ALCAT1 significantly attenuated ROS production, mtDNA depletion, and mtDNA mutation rate in response to oxidative stress. Moreover, ALCAT1 deficiency also prevented mitochondrial fragmentation and swelling by upregulating the expression of MFN2, a mitochondrial GTPase required for the fusion of the inner mitochondrial membrane [18,47].

### 4.2. Fatty Liver Disease

Obesity significantly increases the incidence of nonalcoholic fatty liver disease (NAFLD), yet the underlying connection remains poorly identified. Consistent with its projected role in obesity, ALCAT1 also plays a major role in the pathogenesis of NAFLD. ALCAT1 expression is upregulated in the liver in response to the onset of DIO. Accordingly, targeted deletion of the *ALCAT1* gene prevented the onset NAFLD in response to DIO. Diet-induced NAFLD also caused mitochondrial fragmentation by downregulating the expression of MFN2 [33]. ALCAT1 deficiency prevented mitochondrial fragmentation in hepatocytes, in part by restoring MFN2 expression. Moreover, ALCAT1 deficiency also restored mitophagy, mtDNA copy number, and mtDNA fidelity, leading to significant improvement in mitochondrial respiration in the hepatocytes [33].

### 4.3. Heart Diseases

Heart disease remains the single largest cause of mortality in developed nations. Although major progress has been made in recent years in the prevention and treatment of cardiovascular diseases, there is still a lack of effective treatment for heart failure [71]. The mammalian heart is enriched with mitochondria, and therefore is one of the most sensitive organs to mitochondrial dysfunction [72,73]. ALCAT1 is most abundantly expressed in the heart and plays a key role in cardiomyopathy and heart failure by promoting TLCL deletion. Accordingly, targeted deletion of the *ALCAT1* gene prevented the onset of cardiomyopathy associated with hyperthyroidism and heart failure in mice with myocardial infarction (MI) [17,37]. ALCAT1 deficiency also mitigated myocardial inflammation, fibrosis, and apoptosis in MI mice [17]. In support of key role of ALCAT1 in mitochondrial etiology of heart diseases, ALCAT1 protein expression in the heart was potently upregulated by MI. Consequently, treatment of MI mice with Dafaglitapin (Dafa), a very potent and highly selective ALCAT1 inhibitor, not only restored mitochondrial function but also effectively mitigated MI-induced left ventricle hypertrophy, fibrosis, and contractile dysfunction [17].

CL was initially discovered in the heart and therefore plays a pivotal role in cardiac health [74]. TLCL is the signature CL species in the heart and plays a key role in maintaining normal cardiac function by supporting mitochondrial membrane structure and activities of key enzymes involved in oxidative phosphorylation [15,16]. Consequently, TLCL depletion is implicated in the pathogenesis of various form of heart diseases, including dilated cardiomyopathy, heart failure, and BTHS [11,14,15]. However, the underlying causes for TLCL depletion in heart diseases remain to be identified. Additionally, the onset of heart diseases is associated with pathological remodeling of CL with DHA and other very long chain PUFA [18]. Enrichment of DHA not only renders CL highly sensitive to ROS production and lipid peroxidation in age-related diseases [15,57,75,76,77,78,79], but also leads to the development of cardiac dysfunction by impairing mitochondrial complex I activity during ischemia/reperfusion [11,12]. In further support of ALCAT1 in the mitochondrial etiology of MI, ablation or the pharmacological inhibition of ALCAT1 by Dafa not only restored TLCL levels, but also significantly decreased DHA content in CL in the heart. Consistent with the notion that increased DHA content promotes CL peroxidation, ablation or the inhibition of ALCAT1 by Dafa also prevented ROS production and lipid peroxidation, leading to an increased level of mtDNA copy number sand the restoration of mitochondrial respiration in the hearts of MI mice [17].

As one of the highest oxygen consuming organs in the body, the heart is highly sensitive to damage caused by hypoxia as a consequence of coronary artery blockage. Hypoxia causes oxidative stress, inflammation, and apoptosis of cardiomyocytes, all of which are implicated in the pathogenesis of heart failure. Consistent with predominant expression in microvascular tissues, ALCAT1 also plays an important role in mediating hypoxic response in the heart [64]. Accordingly, the stable overexpression of ALCAT1 in H9c2 cardiomyocytes mimicked the effect of hypoxia by up-regulating the expression of HIF-1α, a key transcription factor and regulator of hypoxic response [17]. Conversely, ALCAT1 deficiency or inhibition by Dafa significantly attenuated HIF-1α protein expression in cardiomyocytes, leading to significant attenuation of myocardial inflammation and apoptosis by decreasing the expression of TXNIP, NLRP3, Bax, and cleaved caspase-3 in the heart. Furthermore, ALCAT1 stimulated HIF-1α protein expression in part through oxidative stress, but not by stabilization of HIF- 1α protein through inhibition of prolyl hydroxylases (PHD) or ubiquitin-proteasome pathways. Accordingly, treatment of cardiomyocytes with Dafa or Mito-Q, a mitochondrial targeted antioxidant, significantly attenuated HIF-1α expression in response to hypoxia. In contrast, the treatment of ALCAT1 deficient cardiomyocytes with either a PHD inhibitor or MG-132 failed to restore the HIF-1α expression level [17]. These findings are corroborated by a previous report that hypoxia promoted HIF-1α stability through ROS production and lipid peroxidation, leading to NLRP3 activation and IL-6 production in the heart [80].

### 4.4. Neurological Diseases

Oxidative stress and mitochondrial dysfunction are implicated in the pathogenesis of neurodegenerative diseases and contribute to the development of Parkinson’s and Alzheimer’s diseases [81,82,83]. In contrast to insulin-sensitive tissues, such as skeletal muscle and the heart, TLCL is not the predominant form of CL species in the brain, representing less than 5% of the total CL [84]. In contrast, CL from a mouse brain is dominated by long-chain PUFA, including DHA (C22:6) and arachidonic acids (C20:4), which contribute to 40% of the acyl side chains of CL [84,85]. Although the biological significance of this acyl composition remains elusive, the high content in PUFA likely increases uncoupling of mitochondrial respiration, leading to increased heat production, which plays an important role in maintaining normal brain function. However, the high content of DHA or other PUFA also renders CL extremely sensitive to oxidative damage in the brain, which may offer an explanation as to why the brain is so sensitive to mitochondrial dysfunction associated with aging. Consequently, traumatic brain injury has been shown to selectively increase the content of DHA in CL and cause CL peroxidation [86]. This selective CL peroxidation preceded the peroxidation of other phospholipids and the onset of apoptosis [86].

Oxidative damage to CL leads to the exposure of oxidized CL to the outer mitochondrial membrane, which serves as a recognition signal for mitophagy [14]. CL externalization to outer mitochondrial membrane is also implicated in the pathogenesis of neurodegenerative diseases. For example, CL externalization to outer mitochondrial membrane leads to mitochondrial dysfunction and α-synuclein aggregation, a key pathogenic event in neurodegenerative diseases [87]. Accordingly, mice with the targeted inactivation of the presynaptic protein α-synuclein exhibited a CL deficiency and a reduction in both TLCL content and the mitochondrial complex I/III activity of the electron transport chain [85]. The knockout mice also exhibited a deficiency of PG, the precursor for CL synthesis. In support of the causative role of ALCAT1 in the pathogenesis of neurodegenerative diseases, ALCAT1 deficiency or inhibition not only prevented MPTP induced Parkinson’s disease but also significantly attenuated the aggregation of α-synuclein, leading to significant improvement in mitochondrial morphology, respiration, and oxidative stress in the brains of a mouse model of Parkinson’s disease [38].

### 4.5. Bath Syndrome

One of the best examples that underscore the importance of CL remodeling in metabolic diseases is BTHS, an X-linked recessive disorder manifested by cardiomyopathy, skeletal muscle myopathy, growth retardation, and neutropenia [88]. BTHS is caused by mutations in the *TAZ* gene, which encodes a transacylase involved in the remodeling of phospholipids [89]. The lipid composition of cells from patients with BTHS shows a dramatic decrease in CL levels and reduced incorporation of linoleic acid (18:2) into CL and its precursor phosphatidylglycerol, even though their biosynthetic capacity to synthesize CL remains unchanged [90,91]. In addition, TLCL, the most predominant CL species in mitochondria from the skeletal and heart muscles, is almost completely absent in BTHS [92]. The mitochondria of BTHS patients exhibit abnormal ultrastructure and respiratory chain defects in the muscles and fibroblasts [88,93,94]. In support of the causative role of ALCAT1 in mitochondrial dysfunction in BTHS, ALCAT1 protein expression in the heart is upregulated by TAZ deficiency, whereas targeted deletion of ALCAT1 not only prevents cardiomyopathy, but also restores mitochondrial morphology and respiration without significant effects on TLCL levels in the hearts of a mouse model of inducible TAZ depletion (Zhang and Shi, unpublished data).

### 4.6. Cancer

Aging is the major risk factor in the development of cancer since the majority of cancer incidences happen in people over the age of 60 [95]. Mitochondrial dysfunction plays a major role in the development of cancer. This notion is supported by the Warburg effect, which is defined as increased glucose uptake and preferential production of lactate [96]. Accordingly, mitochondria are involved in key steps of cancer aggressiveness, tumor growth, and metastasis. In contrast to its projected role in metabolic diseases, little is known about the regulatory function of CL in the development of cancer. Although the dysregulation of CL metabolism has been observed in several types of cancer, much of the evidence remains correlative and ambiguous [97,98]. For example, CL synthesis is implicated in tumor suppression, since mRNA expression level of cardiolipin synthase gene is positively correlated with the survival rate of patients with non-small cell lung cancer (NSCLC) [99]. In contrast, CL remodeling by TAZ has been reported to promote growth and survival of thyroid and cervical cancer cells, suggesting that the role of CL is cancer type-dependent [100,101].

Hypoxia plays a major role in cancer cell metastasis by stimulating the angiogenesis of blood vessels. Consistent with this notion, inhibition of angiogenesis by VEGF inhibitors has been used as a popular treatment for various types of cancers [102]. Consistent with its projected role in age-related diseases, CL remodeling by ALCAT1 potentially controls all aspects of cancer development, growth, and metastasis by promoting Warburg effect, oxidative stress, mitochondrial dysfunction, and angiogenesis. In support of this hypothesis, the overexpression of ALCAT1 led to the Warburg effect, including increased production of lactate and mitochondrial dysfunction [18]. Additionally, ALCAT1 is predominantly expressed in microvascular tissues in mammals and controls the generation of hematopoietic and endothelial lineages in zebrafish [64,103]. Consequently, the inactivation of ALCAT1 in zebrafish prevented the development of blood vessels [103]. Furthermore, ALCAT1 also mediated hypoxic responses, as shown by a recent report [17]. In direct support of the role of ALCAT1 in cancer, a recent report showed that upregulated ALCAT1 expression played a critical role in the development of NSCLC. Accordingly, the ALCAT1 expression level was negatively correlated with the survival rate of NSCLC patients, whereas inhibition of ALCAT1 by a mimetic peptide significantly attenuated cancer cell migration, leading to significant improvement in the survival of mice with NSCLC [65]. However, little is known about the potential role of ALCAT1 in promoting angiogenesis in cancer, which remains to be investigated in future studies.

## 5. Conclusions

The ways in which aging promotes the development of various age-related diseases is one of the frontiers of biological research. Although the mitochondrial free radical aging theory was proposed by Dr. Harman more than 65 years ago, it has become the Holy Grail of aging research. The Holy Grail is traditionally thought to be the cup that Jesus Christ drank from at the Last Supper that holds miraculous powers of eternal youth or sustenance in infinite abundance. Collectively, the information presented in this review provided ample evidence that support a unified theory on CL remodeling by ALCAT1 as the root cause of age-related diseases by controlling the mitochondrial etiology of these disorders (Figure 5). Although much of the evidence accumulated thus far comes from studies on mouse models of age-related diseases, the latest development of potent and selective ALCAT1 inhibitors has made it possible to test this theory in human patients in the near future. However, a key question remains regarding whether the findings can be translated into human clinical studies; if so, ALCAT1 inhibitors could become the first drug that treats all age-related diseases as the Holy Grail. Therefore, a cautionary note must be taken until any human clinical trial results become available in the future. In addition to CL, ALCAT1 also catalyzes the remodeling of phosphatidylinositol and phosphatidylglycerol. Therefore, it can be envisaged that ALCAT1 regulates the development of age-related disorders beyond its role in CL remodeling since both phosphatidylinositol and phosphatidylglycerol play dynamic functional roles in regulating various biological events, such as signal transduction, autophagy, and mitochondrial function. Finally, as a novel mediator of hypoxic responses, the potential role of ALCAT1 in regulating angiogenesis and cancer metastasis remains to be studied in the future.

## Figures and Tables

**Figure 1 cells-11-01906-f001:**
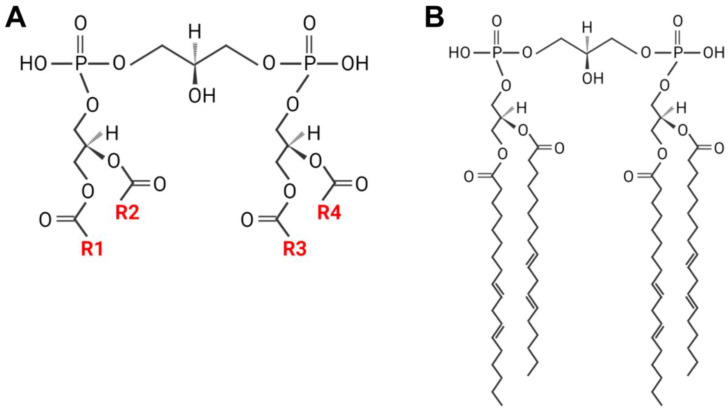
The structure of CL (**A**) and TLCL (**B**). CL has four fatty acyl chains designated R1–R4.

**Figure 2 cells-11-01906-f002:**
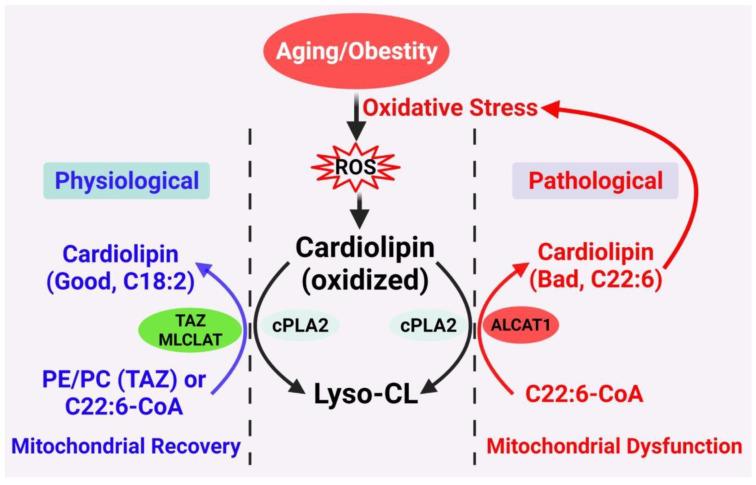
Proposed CL remodeling pathways in obesity and aging. Aging and obesity cause oxidative stress, leading to the production of ROS. CL oxidation by ROS triggers the remodeling of its fatty acyl chains, which begins with hydrolysis of oxidized CL by phospholipase A2 (cPLA2), followed by acylation of lysocardiolipin by either tafazzin (TAZ), MLCLAT, or ALCAT1. TAZ is a transacylase that catalyzes physiological remodeling of CL with other phospholipids, such as PC and PE, whereas MLCLAT catalyzes remodeling of CL with MLCL and linoleoyl-CoA, leading to mitochondrial recovery. ALCAT1 catalyzes remodeling of CL with both MLCL or dilysocardiolipin and docosahexaenoic-CoA (C22:6) as substrates, leading to CL peroxidation by ROS and mitochondrial dysfunction.

**Figure 3 cells-11-01906-f003:**
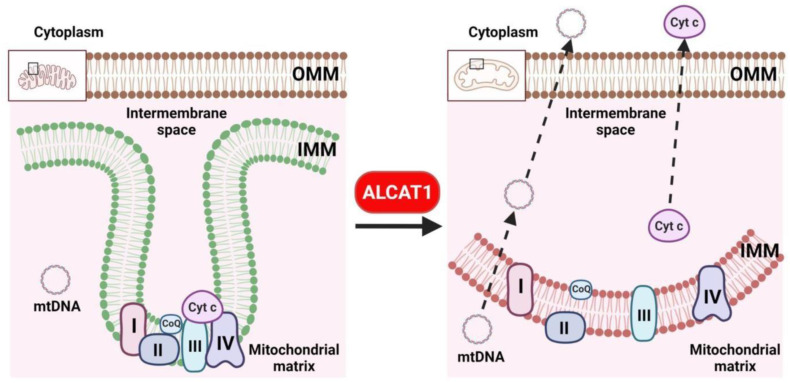
CL remodeling by ALCAT1 leads to a loss mitochondrial membrane curvature, oxidative stress, and defective oxidative phosphorylation. CL is required for the curvature of the inner mitochondrial membrane which plays critical role in supporting the function of mitochondrial respiration complexes. CL remodeling by ALCAT1 causes depletion of CL, leading to a loss of cisterna structure, oxidative stress, defective oxidative phosphorylation, and cytosolic release of mtDNA and cytochrome *c*.

**Figure 4 cells-11-01906-f004:**
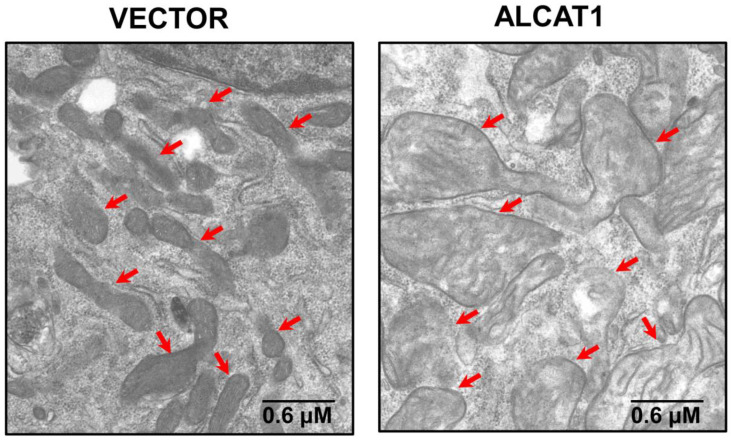
Overexpression of ALCAT1 causes mitochondrial swelling and loss of cristae structure. Electron microscopic analysis of mitochondrial morphology in C2C12 skeletal myocytes cells with stable expression of vector control or ALCAT1 with mitochondria highlighted by arrows.

**Figure 5 cells-11-01906-f005:**
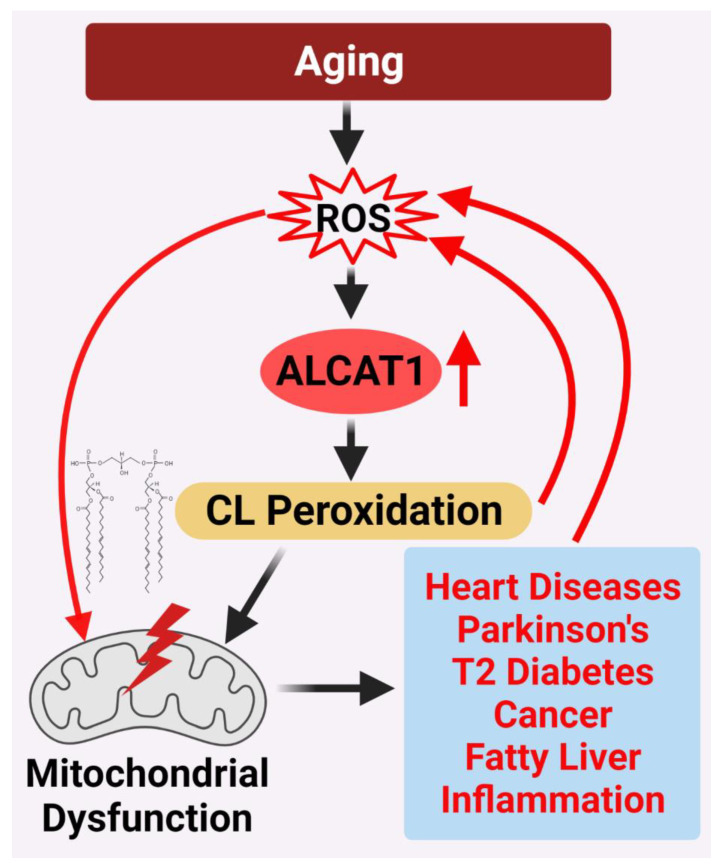
ALCAT1 controls mitochondrial etiology of age-related diseases in response to oxidative stress. ALCAT1 expression is upregulated by ROS from oxidative stress associated with age-related diseases. CL remodeling with PUFA by ALCAT1 causes CL peroxidation, leading to a vicious cycle of ROS production, CL peroxidation, and mitochondrial dysfunction which promote the development of age-related metabolic diseases.

## Data Availability

Not applicable.

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
