# Peer review of "In Search of the Holy Grail: Toward a Unified Hypothesis on Mitochondrial Dysfunction in Age-Related Diseases"

_cells, 2022, doi:10.3390/cells11121906_

Round 1

Reviewer 1 Report

The paper by Jun Zhang  and Yuguang Shi presents an elegant hypothesis that proposed a positive feedback relating  remodeling of cardiolipin wit increased oxidative stress. I think this is a really nice and useful review. It’s difficult to me to estimate the probability that the authors are actually correct, but the logic and data are clearly thought-provoking and educational. I would however, suggest to tone down a bit in declaring the positive feedback they postulate ‘the cause of everything”.

I think attribution images need some attention. For example , the critical Figure 3 has no attribution in the legend and the source and permissions are not clear.

Reviewer 2 Report

This review by Zhang and Shi is a well written review that connects cardiolipin and ALCAT1 to age-related diseases and aging. I have only some minor comments:

 There’s a typo on the first page in after, “* Correspondence; Yuguang Shi,” it should be “Ph.D” and not “h.D.”. Additionally, there are other small typos in the manuscript (many times the space between two words is missing).

 An additional figure with the chemical structure of cardiolipin might be useful.

 Can the authors explain what the usual function of ALCAT1 is under normal conditions? They fully discuss the problems that occur with ALCAT1 overexpression, but there must be some important reason why the body makes this protein. The authors should discuss the normal function of ALCAT1 and its implications for the use of ALCAT1 inhibitors in clinical trials (possible side-effects, etc.).

 There are now several different ways to increase lifespan and healthspan in model organisms. What is known about ALCAT1 levels in these models (e.g. caloric restriction, decreased growth hormone, rapamycin treated mice, etc?)?

 Although the free radical theory of aging is one of the current theories of aging, could the authors discuss how ALCAT1 could fit into other theories of aging (neuroendocrine theory of aging or telomere theory of aging)?

Reviewer 3 Report

Summary. This review explores the possibility that obesity- or age-related “diseases” such as type 2 diabetes, cancer, Alzheimer’s, heart failure etc all have mitochondrial dysfunction as a common denominator. Furthermore it was proposed that peroxidation of cardiolipin is key in causing the age-related mitochondrial dysfunction. The review provides compelling evidence for future investigation of ALCAT1 inhibitors as a treatment for age-related diseases.

Comments. The flow of ideas is logical. However, the m/s needs editing. There were many incidences of words being run together (for example in the title: “MitochondrialDysfunction”), poor English, use of the incorrect tense, miss-use of singular or plural words, and the occasional incorrect word (e.g. Page 2, 2nd line of 2nd paragraph uses “fellow” instead of “follow”).

Figures in general were clear, and assist with clarification of ideas outlined in the text.

Figure 3: The scale bars need to be on both EMs. I assume both images are at the same magnification, but the difference in mitochondrial structure is massive if that is the case! I am used to EMs of isolated cardiomyocytes from rat hearts with clear views of mitochondria, but they are nothing like these images from mice (?) over expressing ALCAT1.
